# IL-1α Processing, Signaling and Its Role in Cancer Progression

**DOI:** 10.3390/cells10010092

**Published:** 2021-01-07

**Authors:** Jing Wen Chiu, Zuhairah Binte Hanafi, Lionel Chin Yong Chew, Yu Mei, Haiyan Liu

**Affiliations:** Immunology Programme, Department of Microbiology and Immunology, Life Sciences Institute, Yong Loo Lin School of Medicine, National University of Singapore, Singapore 117456, Singapore; jingwen.chiu@u.nus.edu (J.W.C.); e0350235@u.nus.edu (Z.B.H.); e0547614@u.nus.edu (L.C.Y.C.)

**Keywords:** interleukin-1α, IL-1R, protease, cancer

## Abstract

Interleukin-1α (IL-1α) is a major alarmin cytokine which triggers and boosts the inflammatory responses. Since its discovery in the 1940s, the structure and bioactivity of IL-1α has been extensively studied and emerged as a vital regulator in inflammation and hematopoiesis. IL-1α is translated as a pro-form with minor bioactivity. The pro-IL-1α can be cleaved by several proteases to generate the N terminal and C terminal form of IL-1α. The C terminal form of IL-1α (mature form) has several folds higher bioactivity compared with its pro-form. IL-1α is a unique cytokine which could localize in the cytosol, membrane, nucleus, as well as being secreted out of the cell. However, the processing mechanism and physiological significance of these differentially localized IL-1α are still largely unknown. Accumulating evidence suggests IL-1α is involved in cancer pathogenesis. The role of IL-1α in cancer development is controversial as it exerts both pro- and anti-tumor roles in different cancer types. Here, we review the recent development in the processing and signaling of IL-1α and summarize the functions of IL-1α in cancer development.

## 1. Introduction

Interleukin-1 (IL-1) was first reported in 1943 when Valy Menkin isolated a pyrogenic euglobulin from inflammatory exudates. This pyrogenic euglobulin, later named as pyrexin, was found to be heat stable and induce biphasic fevers preceded by a marked leukopenia and significant latent period [1,2]. Since then, this substance has been identified and reported by scientists in different research fields and were given different names based on its biological properties, such as endogenous pyrogen, leukocytic endogenous mediator, lymphocyte activating factor, hemopoietin-1, mononuclear cell factor, proteolysis inducing factor, catabolin and osteoclast activating factor [3]. In 1974, Dinarello et al. identified two chemically and biologically distinctive pyrogens derived from monocytes and neutrophils, respectively [4]. Another ten years later, these pyrogens were cloned from both human and mouse and named as Interleukin-1 [5,6]. One year later, Carl J. March et al. successfully isolated these two DNA encoding proteins sharing human IL-1 activity, and named them as IL-1α and IL-1β, respectively [7]. Since then, the function of IL-1β has been extensively studied, while the biological processing and function of IL-1α were less well studied. In this review, we summarize the current understanding of IL-1α processing, signaling, function and its role in cancer progression.

## 2. The Biology of IL-1α

### 2.1. IL-1α Expression

IL-1α is constitutively expressed by many types of cells, including fibroblasts, hepatocytes, keratinocytes, macrophages, dendritic cells, T cells, et cetera [8]. Although various types of cells are known to produce IL-1α, endothelial and epithelial cells are the main source of IL-1α under both physiological and pathological conditions. At steady state, the expression level of IL-1α is relatively low, whereas upon exposure to pathogens or stress-induced stimulus, the expression of IL-1α increases dramatically to initiate inflammatory responses [9].

### 2.2. IL-1α Processing

IL-1α is translated into a 31-kDa precursor form (pro-IL-1α). The pro-IL-1α contains a nuclear localization signal (NLS) site which leads the pro-IL-1α translocation into the nucleus. IL-1α also contains several post-translational modification sites: (1) pro-IL-1α can be phosphorylated at Ser90; (2) pro-IL-1α can be myristoylated at Lys82; and (3) pro-IL-1α can be acetylated at Lys82 [10,11,12,13]. So far, the mechanisms and functions of these post-translational modifications are not well understood. The pro-IL-1α could also be cleaved by several protease to a 17-kDa mature form and a 16-kDa N terminal form named as propiece [14]. The mature form of IL-1α has higher biology activity compared to the pro-IL-1α as its binding affinity to its receptor is several folds higher than the pro-IL-1α [15]. So far, elastase, cathepsin G, proteinase-3, calpain, granzyme B, mast cell chymase, caspase-5, caspase-11 and thrombin have been identified as the pro-IL-1α cleavage proteases [16] (Figure 1).

#### 2.2.1. Elastase, Cathepsin G and Proteinase-3

Neutrophils are the most abundant circulating leucocytes and play a vital role in clearing pathogens through phagocytosis and destruction of microorganisms via granule proteases [17]. The azurophil granule contains three major serine proteases: human leucocyte elastase, cathepsin G and proteinase-3 [18]. Elastase has long been recognized as IL-1β-processing protease and remains unclear whether it could also process IL-1α [19]. In 2011, Afonina et al. first reported that elastase could cleavage pro-IL-1α in vitro when incubating the recombinant IL-1α with elastase. The elastase cleaves IL-1α at the Ala101 site, generating a more biological functional mature form of IL-1α compared with the pro-IL-1α [20]. Recently, the other two neutrophils-derived serine proteases: cathepsin G and proteinase-3 have also been implicated to cleave pro-IL-1α under the in vitro incubation condition. Incubation of pro-IL-1α with cathepsin G and proteinase-3 in vitro generated a smaller processed form of IL-1α compared with that processed by elastase [21]. However, the cleavage site of cathepsin G and proteinase-3 are still unclear. It is of note that all these experiments were performed under in vitro cell-free conditions by incubation of recombinant pro-IL-1α with purified proteases. It is still not clear whether this cleavage occurs in physiological, pathological or both conditions. Supernatants collected from phorbol 12-myristate 13-acetate (PMA)-activated human neutrophils could sufficiently cleave pro-IL-1α, indicating the three serine proteases could process and activate IL-1α under physiological conditions, either alone or at least in combination [21].

#### 2.2.2. Calpain

Calpains are a family of calcium-dependent cysteine proteases, ubiquitously expressed in many types of mammalian cells. Calpains can be directly activated by Ca^2+^ and act as primary second messengers in cellular signal transduction, being involved in many aspects of cellular bioactivity, including cell proliferation, differentiation and programmed cell death [22,23]. In the early 1990s, two independent research groups reported pro-IL-1α served as a substrate for calpain [24,25]. Calpain could cleave human pro-IL-1α between Phe118 and Leu119, whereas murine IL-1α is cleaved between Arg114 and Ser115 [16]. The calpain processed mature form of IL-1α showed several fold increase at inducing IL-2 release from EL4 cells and IL-6 release from vascular smooth muscle cells (VSMCs) compared with the pro-IL-1α, suggesting that the calpain cleaved mature IL-1α has higher bioactivity compared to its pro-form. The possible reason underlying the different bioactivity of mature IL-1α and pro-IL-1α might be due to the different binding affinity to IL-1 receptor, the calpain cleaved mature IL-1α has about 40-fold higher affinity for its receptor compared with the pro-IL-1α [26].

#### 2.2.3. Granzyme B

Granzymes are a family of conserved serine proteases which are well known as cell death–inducing enzymes, mainly stored in NK cells and cytotoxic T cells. Human genes encode 5 granzymes (A, B, H, K, M), while mouse genes encode 11 (A, B, C, D, E, F, G, K, L, M, N) [27]. Granzyme B is one of the most abundant granzymes and has been studied extensively since its discovery in the 1980s [19,28,29,30]. The most recognized role of granzyme B is to induce the death of pathogens infected cells and cancerous cells, thus eliminating the pathogens and preventing cancer initiation and development [31]. In 2011, Afonina et al. first found that IL-1α was the substrate for granzyme B, while IL-1β was not. Granzyme B could cleave IL-1α at Asp103, a conserved motif among mammals [20]. It is worth noting that although perforin is essential for the cytotoxicity function of granzyme B, more and more evidences have shown that besides NK and T cells, non-cytotoxicity cells such as keratinocytes, mast cells and breast cancer cells also express granzyme B while no perforin is expressed [32], suggesting granzyme B alone can function as an extracellular protease to cleave pro-IL-1α released from the dead cells to mature IL-1α, thus boosting the inflammatory responses [33].

#### 2.2.4. Chymase

Chymase is a mast cell specific protease that belongs to the family of serine proteases and is involved in many pathological conditions [34]. Chymase was first explored as the protease against IL-1β in 1990s [35]. In 2011, Afonina et al. started to investigate the possibility of whether chymase could also cleave IL-1α and found that chymase cleaved pro-IL-1α at Phe116. Chymase is largely released during degranulation of mast cells, easily to meet and cleave the pro-IL-1α leaked from dead cells to enhance the bioactivity of IL-1α [36,37].

#### 2.2.5. Caspase-5 and Caspase-11

Caspases are a family of cysteine proteases and are inseparably linked to programmed cell death. Hundreds to thousands of potential caspase substrates have been identified in multiple biological processes, such as cell apoptosis, differentiation, tissue repair as well as tumorigenesis [38]. In 2019, Kimberley et al. found that murine caspase-11 and its human orthologue caspase-5 served as IL-1α protease [39]. During noncanonical inflammasome activation, murine caspase-11 cleaves pro-IL-1α at Asp106 site, resulting in increased IL-1α activity. Human caspase-5 shared the same cleavage site with granzyme B at Asp103. Both caspase-5 and caspase-11 are required for the senescence-associated secretory phenotype to release IL-1α during cell aging, thus targeting caspase-5 or caspase-11 might be a potential therapeutic strategy to reduce senescence-associated inflammation.

#### 2.2.6. Thrombin

Thrombin, also known as coagulation factor II, is a serine protease that plays a vital role in maintaining blood coagulation and regulating homeostasis [40]. Various studies suggest thrombin is also involved in cell signaling and inflammation. In the early 21st century, Antonella et al. first reported the relationship between thrombin and IL-1. They found that thrombin enhanced the release of both IL-1α and IL-1β in phytohemagglutin treated human peripheral blood mononuclear cells (PBMCs). Further studies showed that thrombin increased IL-1 production at both mRNA and protein level in PBMCs. Seventeen years later, Laura et al. explored the potential proteolytic role of thrombin in processing pro-IL-1α. They found that mammalian pro-IL-1α contains a highly conserved proline-rich sequence(PRS) motif which can be cleaved by thrombin to a fragment of ~18-kDa (p18). The p18 fragment had similar activity compared with calpain cleaved p17 fragment [41]. On one hand, the cleaved p18 initiates and boosts the local immune responses; on the other hand, it feeds back to hemostasis by boosting thrombopoiesis. This finding has filled a scientific gap of how coagulation is directly linked to innate immune response.

### 2.3. IL-1α Signaling

IL-1 signaling pathway is triggered after binding of IL-1α to the IL-1 receptor type I (IL-1RI). IL-1RI is found on the surface of various types of cells and, both IL-1α and IL-1β act on this same receptor. Its structure consists of intracellular domain; known as the Toll-like/IL-1R (TIR), which has similar homology with other members of IL-1R and TLR families, as well as three extracellular immunoglobulin domains [42]. Binding of IL-1α to its receptor leads to dimerization of the receptor at the TIR domain and recruitment of IL-1R accessory protein, IL-1RAcP [43]. This complex then recruits other adaptor protein, myeloid differentiation factor 88 (MyD88) and IL-1R-associated kinase (IRAK). Formation of this signaling module: IL-1α, IL-1R1, IL-RAcP, MyD88 and IRAK4, is essential for subsequent downstream signaling cascade. Mice lacking the IRAK4 gene were shown to be incapable of eliciting IL-1 signaling response whereby productions of tumor necrosis factor-α (TNF-α) and IL-6 were completely abolished in the knockout mice [44]. In addition, binding of nuclear factor kappa-B (NF-κB) to DNA motif as well as activity of stress kinases such as c-Jun N-terminal kinases (JNK) and p38 mitogen activated protein kinases (MAPK) were reduced in IRAK4^−/−^ macrophages cells after stimulation with lipopolysaccharides (LPS), suggesting that IRAK4 is a key molecule needed for IL-1 signaling [44,45]. Furthermore, serum of mice with MyD88 deficiency have significantly lower levels of TNF-α and IL-6 as compared to the wild-type mice after being intravenously injected with recombinant IL-1β [46]. Therefore, formation of MyD88 and IRAK4 complex is a critical mediator in IL-1 signaling cascades. Assembly of this signaling module results in phosphorylation of IRAK4 which in turn phosphorylates IRAK1 and IRAK2 [45,47]. Tumor necrosis factor–associated factor (TRAF) 6 is then recruited and associates with the complex. IRAK1-IRAK2-TRAF6 complex dissociates and migrates to the plasma membrane to associate with transforming growth factor-β-activated kinase (TAK1) and its binding partners; TAK1-binding proteins TAB1 and TAB2. TAK1-TAB1-TAB2-TRAF6 complex then translocates to the cytoplasm where TAK1 is phosphorylated, hence dissociating itself from the complex [48].

After dissociation from the complex, TAK1 has the ability to initiate two main signaling pathways. The first is the activation of NF-κB pathway. The core IKK complex is made up of IKKα and IKKβ together with the regulatory subunit NF-κB essential modulator (NEMO). In this case, NEMO binds to the polyubiquitin chain of IRAK1 while TAK1 binds to either TAB2 or TAB3. Inhibitor of NF-κB kinase subunit beta (IKKβ) can be activated by TAK1 which in turn phosphorylates IκB on specific serine residues, promoting the K48-linked polyubiquitination [47,49]. This results in degradation of IκB by proteasome, allowing the release of NF-κB (p50 and p65), its translocation into the nucleus and expression of important proinflammatory cytokines genes such as IL-6, TNF-α and cyclooxygenase 2 [47,48]. Another pathway is the MAPK pathway. This involves the activation of p38 MAPK, JNK and extracellular signal-regulated kinases (ERK) through interaction of TAK1 with the MKKs. The p38 MAPK and JNK are the two pathways that are activated during stress conditions that are mainly linked to apoptosis while ERK is associated with cell survival [50]. p38 MAPK and JNK are considered to have tumor suppressive roles while ERK is more towards tumor promoting. For instance, this activation of stress induced MAPK could exert its tumor suppressive role through increased production of proapoptotic proteins, Bax and Bim, and decreased production of anti-apoptotic proteins, Bcl2, hence promoting apoptosis [48].

Like other signaling pathways, IL-1 signaling cascade is under strict regulation. For example, the presence of IL-1RII acts as a decoy receptor which could bind to pro-IL1α, thus preventing it from being cleaved by proteases such as calpain [26,43]. IL-1Ra is a receptor antagonist that binds to IL-1R1, hence inhibiting its activation [51]. Another regulatory pathway is through binding of IL-1R1 to the adaptor toll-interacting protein (TOLLIP), inhibiting IRAK1 thus resulting in internalization of IL-1R1 to endosomes for degradation [52]. Several negative feedback mechanisms are also present such as the production of IκBα preventing NF-κB signaling and TAB1 phosphorylation by p38 MAPK inactivating TAK1 activity [47,53] (Figure 2).

### 2.4. IL-1α Cellular Localization

#### 2.4.1. Nucleus Localization

Pro-IL-1α contains a NLS (KVLKKRRL) located at the N terminal, in front of all the known proteases cleavage sites [54,55]. Both pro-IL-1α and the cleaved N-terminal fragment can translocate into the nucleus in epithelial cells, myeloid cells, fibroblasts and keratinocytes [55]; while in VSMCs, the NLS site of pro-IL-1α is shielded by IL-1RII, preventing IL-1α nucleus translocation and retaining the IL-1α in the cytosol [26]. The underlying mechanisms on how IL-1α is translocated into the nucleus are still unclear. Cohen et al. reported that acetylation at Lys82 helped the nucleus translocation of pro-IL-1α in mouse macrophage cell line Raw 264.7 cells under genotoxic stress condition [13]. Under steady state, pro-IL-1α nuclear translocation is regulated by HS-1-associated protein X-1 (HAX-1). HAX-1 is a ubiquitously expressed protein and predominantly located at mitochondrial and endoplasmic reticulum membrane as well the nuclear envelope [56]. HAX-1 could bind to pro- and N-terminal IL-1α at three potential sites and facilitate their nuclear translocation [57,58]. Depletion of HAX-1 expression using siRNA abolished the nucleus translocation of IL-1α, demonstrating the requirement of HAX-1 for IL-1α nucleus translocation.

The proteins and chromatin interacting with nucleus localized IL-1α are not well studied. In vitro overexpression systems revealed that N-terminal IL-1α physically interacts with mammalian histone acetyltransferases Gcn5, PCAF and p300, and with adaptor protein Ada3 [45]. Confocal microscopy also showed that endogenous pro-IL-1α colocalized with histone H3 in LPS-stimulated macrophages [42]. We previously reported N-terminal IL-1α could bind to the promoter of specific protein 1 (SP1) and promoted the growth of human T-cell acute lymphoblastic leukemia cell line [43]. Yeast two-hybrid interaction analysis showed that N-terminal IL-1α interacted with five proteins ASF/SF2, CC1.4, hnRNP A/B, SMA5 and prp8, which are all involved in RNA processing, thus promoting various malignant cells apoptosis [44].

#### 2.4.2. Membrane IL-1α

In 1985, Evelyn A et al. found fixed macrophage monolayers and isolated membranes from unfixed macrophages promoted T cell proliferation, indicating a membrane bound form of IL-1α existed [46]. This finding was later confirmed by many studies that the pro-IL-1α was anchored on the cell membrane and functioned through juxtracrine and paracrine manner [9,47,48]. It is still not clear how the cytosol pro-IL-1α translocated onto the membrane. D.T. Brody et al. proved that membrane-bound pro-IL-1α is glycosylated, indicating post-translational modification is required for pro-IL-1α membrane localization and lectin is involved in the binding [59]. We previously reported that in tumor cells, mutation of both NLS and calpain cleavage site led to increased membrane localization of pro-IL-1α, suggesting the NLS signal and protease cleavage reduced pro-IL-1α membrane translocation [60]. Furthermore, membrane-bound pro-IL-1α was also found to be cleaved by extracellular protease into mature form and function through the IL-1R signaling [61].

#### 2.4.3. Cytosolic IL-1α

IL-1α protein was translated in the cytosol and then translocated into the nucleus or membrane. However, confocal images also showed that IL-1α was located in the cytosol. IL1-RII acts as a bona fide decoy receptor of IL-1 and serves as a potential mechanism to retain pro-IL-1α in the cytosol [62,63]. IL1-RII is highly conserved and functions to dampen inflammation caused by IL-1 [64]. Furthermore, IL1-RII binds to IL-1α with a high affinity (Kd ≈ 10^−8^ M) but binds IL-1Ra with a much lower affinity which allows IL-1RII to effectively inhibit the functions of IL-1α [65]. IL-1RII in its full length can act as a membrane bound decoy receptor and prevents any downstream signaling upon binding to IL-1α which would otherwise normally initiate a pro-inflammatory response [66]. It can also be proteolytically processed by metalloproteinase, a disintegrin and metalloprotease 17 (ADAM17), and eventually be released in a soluble form (sIL-1RII) that stays in the cytosol of the cell [58,67]. sIL-1RII had been shown to interact with IL-1α, specifically in the pro-IL-1α form, and that the binding of sIL-1RII to pro-IL-1α prevents the processing and cleavage of pro-IL-1α into its mature/secreted form by several enzymes such as calpain and granzyme B [20], thereby dampening IL-1α activity. This process could be reversed by caspase 1, which is able to cleave intracellular sIL-1RII thus preventing it from binding to pro-IL-1α [26]. The activity of IL-1RII to reduce IL-1-induced inflammation had been exhibited in several different pathological conditions in vivo. In a model of arthritis in rabbits, intravenous introduction of soluble IL-1RII was shown to reduce arthritis symptoms such as lessening of joint swelling [68]. In a model of heart transplant in mice, introduction of IL-1RII was shown to reduce inflammation by observing a decreased in infiltrating macrophages and CD4^+^ T cells and also lowering of pro-inflammatory cytokines such as TNF-α and TGF-β [69]. Thus, IL-1RII, being able to mitigate IL-1α signaling via multiple complimentary pathways, makes it a robust anti-inflammatory agent.

#### 2.4.4. Secreted IL-1α

IL-1α lacks the signal peptide, therefore it is secreted through unconventional pathways. For a long time, the secretion of IL-1α was considered as passive release due to the loss of cell membrane integrity, such as necrosis or physical damage [9]. However, accumulating evidences suggest that IL-1α may be secreted through multiple regulatory pathways. In 1995, Keisuke found monocytes derived from the interleukin-1β converting enzyme (caspase-1) KO mice did not export either IL-1β or IL-1α upon LPS stimulation, indicating caspase-1 involved in IL-1α secretion [41]. Moreover, caspase-1-dependent inflammasomes-mediated IL-1β maturation and release also induce the IL-1α release in macrophages [14,70]. Though caspase-1 is required for IL-1α secretion, the export route of IL-1α from cytosol to extracellular space is still unclear. One hypothesis is IL-1α hijacked IL-1β for section, as IL-1β–deficient bone marrow derived dendritic cells (BMDCs) were unable to secrete IL-1α and in vitro overexpression experiments proved an interaction between IL-1α and IL-1β [54]. Another hypothesis is that IL-1α secretion is dependent on Cu^2+^ binding protein known as S100A13 [71]. Indeed, a study performed by Mandinova et al. showed that in human U937 cells, S100A13 is associated intracellularly with IL-1α in the context of IL-1α exportation when the cells underwent heat-stress [72]. Therefore, the exportation of IL-1α might have some similarities to that of relatively well-studied fibroblast growth factor 1 (FGF1), whose exportation has been shown to be associated with S100A13 [71]. Interestingly, IL-1 prototypes and FGF gene family exhibit high levels of structural homology [73]. Exportation of IL-1α, similar to FGF1, might occur via a Cu^2+^ -dependent formation of multiprotein complexes involving S100A13 which are translocated across the membrane as a molten globule [74].

Pyroptosis, a newly identified gasdermin-mediated programmed cell death, is found to be involved in IL-1β secretion in inflammatory cells, such as macrophages and neutrophils [13,57,58]. Pyroptosis is mainly trigged by danger signals and activates the caspases. Activated caspase-1 and caspase-11/4/5 cleave gasdermin D (GSDMD) in inflammatory cells, while caspase-3 cleaves gasdermin E (GSDME) in tumor cells [58], to release the N terminal domain of gasdermins that inserts into the plasma membrane to form pores that allow intracellular molecules to pass through. Though most of the pyroptosis studies have been focused on the inflammasome formation and IL-1β secretion, a few pieces of evidence suggest that IL-1α may also take advantage of gasdermin pores to be secreted out. Samantha et al. recently found that in a murine *Toxoplasma gondii* infection model, microglia, not peripheral myeloid cells, were the main source of IL-1α. The section of IL-1α in microglia was diminished in the GSDMD or caspase-1/11 deficient mice, suggesting GSDMD serves as the export route for IL-1α [75].

Recently, a universal unconventional protein secretion (UPS) route for leaderless protein was discovered. The chaperone HSP90A binds and unfolds leaderless proteins, including mature IL-1β, IL-1α, IL-33, Tau, et cetera. This HSP90A-bound cargo then interacts with the TMED10, resulting in the oligomerization of TMED10 to form a protein channel that translocates the leaderless protein into the lumen of the ER-Golgi intermediate compartment (ERGIC) for subsequent secretion out of the cell via a vesicular intermediate [76].

To our knowledge, IL-1α is the only cytokine that locates in nuclear, cytosol, plasma membrane as well as being secreted. The underlying mechanisms that determine and regulate the IL-1α subcellular localization remain a major question in this field [77] (Figure 3).

## 3. IL-1α in Cancer Development

Inflammation is closely related to carcinogenesis [78]. As an important pro-inflammatory cytokine, the role of IL-1α in cancer growth has been studied in various cancer types, and the results suggested IL-1α is a dual function cytokine which exerts pro- and anti-tumor function in different cancers. Here, we summarized the recent progress for the role of IL-1α in multiple cancer types and the possible mechanisms of how IL-1α promoted or inhibited tumor growth (Table 1).

### 3.1. Breast Cancer

In a study with MCF-7 breast cancer cells overexpressing mature IL-1α, when transplanted into nude mice, the MCF-7 cells swiftly formed fast growing estrogen-dependent tumors [49]. However, an earlier study by another group showed that IL-1α was able to inhibit MCF-7 breast cancer cell growth at the G0/G1 phase of the cell cycle [50]. The conflicting results of these studies could be due to their different experimental systems as Sgagias et al., exposed the MCF-7 cells to exogenous IL-1α in an in vitro setting, while Kumar et al. overexpressed IL-1α in the MCF-7 cells and studied the effects in vivo. While IL-1α expression alone in MCF-7 breast cancer cells was not able to induce metastasis, a study on breast tumor-derived IL-1α resulted in metastatic spread by inducing the expression of thymic stromal lympopoietin (TSLP) from tumor-infiltrating myeloid cells [80]. TSLP is a critical tumor survival factor required for metastasis and it promotes survival through induction of Bcl-2 expression [80]. This study introduced a novel IL-1α-TSLP-mediated crosstalk between breast tumor cells and tumor-infiltrating myeloid cells regulating metastatic breast cancer.

HER2 is overexpressed in 25% of all breast cancers, and has been correlated with poor prognosis, high relapse rate, and aggressive symptoms [51]. Hence, a study looked into the potential relationship between HER2 expression, inflammation and cancer stem cell (CSC) expansion [52]. Here, they found that HER2 induces expression of IL-1α and IL-6 in a positive feed-forward activation loop. In turn, activation of NF-kB and signal transducer and activator of transcription 3 (STAT3) were increased, leading to CSC generation and maintenance. Blockade or absence of IL-1α signaling also led to delayed tumorigenesis, reduced CSC population in tumors, reduced inflammation and improved chemotherapeutic efficacy. Another study also showed that high expression of IL-1α in breast cancer biopsies was associated with tumor dedifferentiation and more malignant phenotypic behavior [53], hence supporting the previous study of IL-1α’s role in tumorigenesis and potential CSC generation and maintenance. Conversely, a study on mammary tumor cell proliferation in MMTV-PyMT breast cancer cell showed that IL-1α dependent IL-1R signaling could suppress tumor growth and curb pulmonary metastasis [59].

### 3.2. Pancreatic Cancer

IL-1α expression by metastatic pancreas cell lines was able to induce hepatocyte growth factor (HGF) secretion by fibroblast [60]. HGF will then enhance pancreatic cancer invasion, proliferation and angiogenesis. Similarly, IL-1α secretion by MiaPaCa-2 cells causes NF-kB to be constitutively activated, resulting in NF-kB downstream genes that are involved in metastatic cascade and angiogenesis to be expressed [61]. In another study, coculture of cancer associated fibroblasts (CAFs) with pancreatic ductal adenocarcinoma cells (PDAC) resulted in upregulation of IL-1α and other inflammatory factors, creating an inflammatory tumor microenvironment suitable for tumor survival and progression [77]. Moreover, IL-1α was found to be the key in the formation and maintenance of this inflammatory tumor microenvironment as blockage of IL-1α alone significantly ablated the production of other inflammatory factors. Hence, in the context of pancreatic cancer, IL-1α expression in the presence of tumor-associated fibroblasts seemed to synergistically promote tumor growth and progression.

### 3.3. Leukemia

When acute lymphocytic leukemia (ALL) cells were stimulated with mature IL-1α, they proliferated in a dose-dependent manner. Analysis of patient samples also saw spontaneous expression of IL-1α in ALL cells that usually do not express IL-1α [71]. Thus, this strongly suggests that IL-1α could act as an autocrine growth factor in ALL cell proliferation. Overexpression of the propiece form of IL-1α increased the growth of ALL cells through the activation of NF-kB and SP1. In addition, the propiece-expressing ALL cells also had reduced apoptosis and were more resistant to low serum concentration and cisplatin treatment [43]. Interestingly, different forms of IL-1α in these studies resulted in the same effect of increased ALL proliferation. It is possible that both forms activated similar pathways, as the mature IL-1α used by Mori et al. could have bound to IL-1R1 on ALL cells and signaled to increase the expression of NF-kB; while the propiece IL-1α used by Zhang et al. directly activated expression of NF-kB in the nucleus of ALL cells.

When adult T-cell leukemia (ATL) cells from patients were cultured with recombinant IL-1α, it also proliferated in a dose dependent manner [86]. In another study looking at ATL cells from patients, some were able to release biologically active IL-1α protein [87]. The results of these two studies are similar to that in ALL cells [71]. Hence, supporting the idea that IL-1α could act via an autocrine mechanism to stimulate growth and also play a role in the pathological changes associated with ATL. However, conversely when malignant T lymphoma cells were stimulated in vitro to transiently expressed IL-1α, they showed antitumor and immunotherapeutic effects in vivo [88].

### 3.4. Ovarian Cancer

It has been found that a missense single nucleotide polymorphism (SNP) in IL-1α leads to the full length intracellular IL-1α to be more readily cleaved by inflammatory proteases [101]. In a very large sample size of nearly 40,000 participants, this A114S SNP in IL-1α (rs17561), was also found to be associated with a higher risk of ovarian cancer [85]. Thus, this suggests that increased IL-1α secretion could result in increased tumorigenicity and metastasis in ovarian cancer.

### 3.5. Head and Neck Squamous Carcinoma

Head and neck squamous carcinoma (HNSCC) patients with distant metastasis had higher IL-1α expression compared to those without metastasis [81]. Patients with higher IL-1α HNSCC were also found to have significantly lower five-year distant metastasis-free survival compared to those with low IL-1α HNSCC. Hence, IL-1α expression could be a prognostic marker of distant metastasis in HNSCC patients. Conversely, when used as a treatment, IL-1α in combination with existing EGFR inhibitor therapy, cetuximab, was found to effectively induce T cell dependent anti-tumor immune response [82]. Patients with higher baseline IL-1α expression and treated with cetuximab was also associated with more favorable progression free survival. This could potentially point towards the use of IL-1α as an immunotherapy option for EGFR-positive HNSCC.

### 3.6. Liver Cancer

The risk of hepatocellular carcinoma (HCC) is known to increase with chronic liver injury and inflammation. A study using a model of diethylnitrosamine (DEN)-induced liver carcinogenesis in mice found that IL-1α released from hepatocyte necrosis resulted in inflammation and a compensatory proliferative response to regenerate the liver, thus contributing to the development of HCC [84]. Inhibition of IL-1R1- or IL-1α-mediated signaling pathways could prevent HCC development. Thus, IL-1α is a key driver of HCC development in this mouse liver injury model. Meanwhile, another study looking at the function of the membrane IL-1α in murine HCC models found potential inhibitory functions on tumor development via immune cell activation [60]. In the mice models, membrane IL-1α was able to activate anti-tumor immunity via promoting CD8^+^ T- and NK-cells functions against the malignant cells.

### 3.7. Lung Cancer

Highly metastatic human lung cancer cell line was found to express higher levels of IL-1α, compared to their lower metastatic counterparts, and was able to form tumors with enhanced angiogenesis and lymphangiogenesis [79]. The tumors were also found to be extensively infiltrated by M2-type macrophages. Since these effects could be suppressed by IL-1R antagonist, the activation of angiogenesis and lymphangiogenesis seems to be IL-1α driven.

### 3.8. Fibrosarcoma

In a 3-MCA-induced fibrosarcoma cell line, IL-1α expression was required for tumor development when implanted into mice [89]. The authors also found that IL-1α played an important role in controlling immune-surveillance of the developing tumor [89]. Conversely, when studying the cytosolic full length and membrane form of IL-1α on fibrosarcoma, many other studies show a suppressive effect on tumor growth. In an oncogene-transformed NIH/3Y3-derived cell line overexpressing cytosolic or membrane-bound forms of IL-1α, IL-1α exerts adjuvant-like effects, increasing the immunogenicity of tumor-cell antigens and regression of the tumor [90]. Another group investigated the mechanism of this tumor suppressive effect and found that the IL-1α activated the development of tumor cell-specific cytotoxic T lymphocytes (CTL). These CTLs were important for tumor eradication, and the authors also found that fibrosarcomas that do not produce cytosolic or membrane-bound IL-1α can be induced to do so via treatment with cytokines or LPS. After induction, the tumor started regressing and mice were also able to develop immunological memory that protected them from re-challenge [91]. Another more recent study on cytosolic or membrane-bound IL-1α in fibrosarcoma also supported the above claims. The authors found that at the injection sites of IL-1α positive fibrosarcoma cells, CD 8^+^ T-cells, NK-cells and macrophages accumulated there [92]. The tumor regressed and was replaced by fibrotic scar tissue. It was hypothesized that the membrane-associated IL-1α on the fibrosarcomas may be able to interact with immune effector cells bearing IL-1R, thus activating them and inducing the anti-tumor activities. Overall, cytosolic or membrane-bound IL-1α in fibrosarcoma seems to induce anti-tumor activity via activation of immune cells.

### 3.9. Gastric Cancer

Similar to HNSCC, there is also a correlation between distant liver metastasis in gastric carcinoma patients and higher levels of IL-1α expression [83]. In patient samples, differentiated tumors were more likely to stain positive for IL-1α compared to undifferentiated tumors. In addition, there is a significantly higher incidence of liver recurrence in patients with tumors expressing IL-1α compared to those without, hence, again showing a correlation between IL-1α expression in tumor and metastasis.

In other studies of human gastric cancers, it was found that expression of mature IL-1α was significantly correlated with the liver metastatic potential and angiogenesis of gastric cancer [93]. In coculture of IL-1α-expressing gastric cancer cells and human umbilical vein endothelial cells (HUVEC), HUVEC strongly enhanced proliferation and tube formation which could be inhibited by blockade of IL-1α, thus suggesting IL-1α being the key mediator in the proliferation and angiogenesis. In another study, Sakamoto et al. found that mice treated with a DNA damaging agent, N-methyl-*N*-nitrosourea (MNU), increased the expression of IL-1α in gastric epithelial cells (GECs), and was positively correlated with cancer progression. Additionally, they showed that the IL-1α upregulation was via IKKβ signaling, and that it conferred the pre-malignant cells anti-apoptotic and cell proliferation effects, which then eventually led to dysplasia [95]. Hence, IL-1α is a key mediator in the proliferation and tumor progression in gastric cancers.

In another study, Shibata et al. found that IKKβ conditional knockout mice showed similar results when exposed to stress such as ionizing irradiation or *Helicobacter felis* infection [94]. However, in this model, IL-1α was released from the GECs due to necrosis, as opposed to being secreted out by GECs as shown in the previous study. The release of IL-1α resulted in the recruitment of inflammatory cells that accelerated the progression to gastric pre-neoplasia. Thus, given all the evidence in gastric cancer, IL-1α seems to play a promoting role in gastric cancer progression.

### 3.10. Prostate Cancer

Analysis of samples from patients with prostate cancer showed that IL-1α expression in malignant cells is associated with the increased progression of the disease, as well as in correlation with prostate specific antigen (PSA) levels in the serum [96]. Cytokine profiling identified IL-1α and IL-6 originating from prostate epithelial cells to act synergistically in driving PSA expression. The authors also hypothesized that IL-1α might drive PSA expression via NF-kB activation.

### 3.11. Oral Squamous Cell Carcinoma

Oral squamous cell carcinoma (OSCC) cells and CAFs were found to interact together to establish an inflammatory tumor microenvironment that contributes towards carcinogenesis [97]. OSCC cells can release IL-1α that stimulates CAF proliferation, concomitantly CAF increases the secretion of CCL7, CXCL1, and IL-8 cytokines that help facilitate cancer progression and invasion.

### 3.12. Cervical Cancer

In human papilloma virus (HPV)-induced cervical cancer cells, IL-1α was found to act as an adjuvant by activating dendritic cells for anti-tumor effects [102]. Hence, the authors used viral dsRNA (polyIC) to induce RIPK3-dependent necroptosis in cervical cancer cells, resulting in the release of intracellular IL-1α, which could then activate dendritic cells to produce IL-12 and mount an anti-tumor response.

### 3.13. Skin Cancer

To investigate the role of IL-1α and inflammation in cutaneous cancer progression, FVB/N transgenic mice were generated overexpressing the mature IL-1α in the epidermis. Interestingly, the authors found that the mice were resistant to carcinomas that required evolution from prior papillomas, but carcinomas still developed de novo at an accelerated rate compared to the controls [98]. In another study, A375 human melanoma cells were transfected to overexpress and secrete mature IL-1α constitutively, then they were intravenously injected into mice. This resulted in potent inflammation and induced adhesion molecule expression on lung endothelial cells (EC), enhancing their adhesiveness to tumor cells, thus resulting in increased tumor-cell retention in the lungs [99].

### 3.14. Kidney Cancer

Glomerular mesangial cells, a model perivascular myofibroblast cell that is able to synthesize and process IL-1α, was transduced to overexpress the propiece form of IL-1α. This resulted in the malignant transformation of the cells into a spindle cell-type tumor [75].

## 4. Further Perspectives

IL-1α is a unique cytokine, unlike its homologue IL-1β, both the pro- and mature forms are biologically functional. Besides, the propiece IL-1α has also been implicated as a functional regulator in cell growth and immune responses. Moreover, IL-1α could (1) translocate into the nucleus to regulate downstream gene expressions independent of the IL-1R signaling, (2) anchor to the cell membrane to activate IL-1/IL-1R signaling, (3) secrete out of the cell and activate the IL-1/IL1R signaling, and (4) stay in the cytosol with unknown function. It is of great importance to understand the mechanisms and functions of these different IL-1α isoforms in different inflammatory diseases as well as cancers. This multifaceted aspects of IL-1α functions implicated that it is critical to target IL-1α isoforms precisely for immunotherapy, though anakinra (Kineret; Amgen/Biovitrum), the IL-1R antagonist protein has already been approved by the FDA and has shown promising results for treating patients with rheumatoid arthritis and auto-immune diseases [103,104]. Besides, anakinra blocks both IL-1α/IL-1R and IL-1β/IL-1R signaling, which may have differential or opposite functions in certain disease conditions. Moreover, IL-1α could exert its function in an IL-1R independent manner which is not affected by anakinra. Thus, it is crucial to reveal the detailed roles of IL-1α isoforms in different cancer types as well as the mechanisms of their processing and regulation in order to discover new drug candidates that can target IL-1α functions more precisely for future immunotherapy.

## Figures and Tables

**Figure 1 cells-10-00092-f001:**
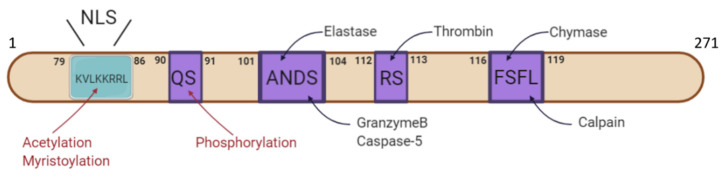
Overview of IL-1α structure. Human IL-1α consists of 271 amino acids including an nuclear localization signal (NLS) domain. IL-1α can be cleaved by various proteases, such as elastase, granzyme B, caspase-5, thrombin, chymase and calpain. IL-1α also undergoes post-translational modification, including acetylation, myristoylation as well as phosphorylation.

**Figure 2 cells-10-00092-f002:**
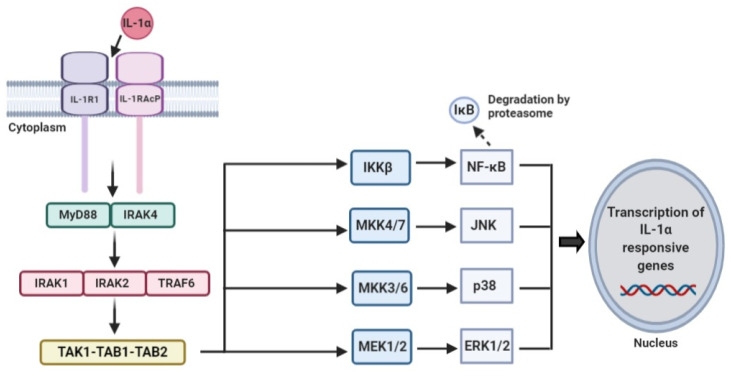
Overview of IL-1α signaling pathway. The binding of IL-1α to IL-1RI leads to dimerization of the receptor at the TIR domain and recruitment of IL-1RAcP. This IL-1α/IL-1RI/IL-1RAcP complex then recruits adaptor proteins MyD88 and IRAK for subsequent downstream signaling cascade. IRAK will phosphorylate IRAK1 and IRAK2 and recruit TRAF6 to form the IRAK1-IRAK2-TRAF6 complex; subsequently TRAF6 will associate with TAK1-TAB1-TAB2 to phosphorylate TAK1. Phosphorylated TAK1 will dissociate from the TRAF6-TAK1-TAB1-TAB2 complex to initiate the NF-κB and MAPK signaling pathways to regulate the downstream proinflammatory cytokines expressions, such as IL-6, TNF-α and cyclooxygenase 2. Abbreviations: IL-1RI, IL-1 receptor I; IL-1RAcP, interleukin-1 receptor accessory protein; MyD88, myeloid differentiation factor 88; IRAK, IL-1R-associated kinase; TRAF6, tumor necrosis factor–associated factor 6; TAK1, transforming growth factor-β-activated kinase 1; NF- κB, nuclear factor kappa-B; MAPK, Mitogen-activated protein kinase.

**Figure 3 cells-10-00092-f003:**
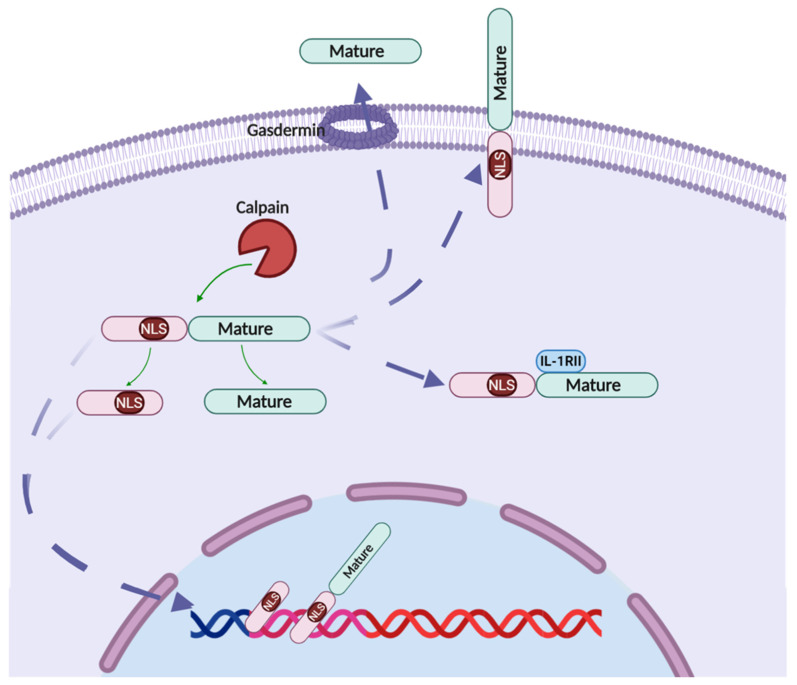
Overview of IL-1α cellular localization. IL-1α is translated as pro-peptide precursors form. (1) Pro-IL-1α contains a nuclear localization sequence (KVLKKRRL) in the N-terminal region which leads pro-IL-1α and propiece IL-1α translocation into the nucleus; (2) Pro-IL-1α can be cleaved by several proteases, such as calpain, into mature form. Mature IL-1α can be secreted out of cell possibly through gasdermin-formed pores; (3) Pro-IL-1α can be myristoylated and anchored to the cell membrane to form the membrane-bound IL-1α; (4) IL-1RII binds to Pro-IL-1α, shielding the NLS and proteases cleaved sits, thus retaining the Pro-IL-1α in the cytosol.

**Table 1 cells-10-00092-t001:** Summary of IL-1α function in tumor development.

Tumor Type	Tumorigenesis	Mechanism of Action	IL-1α Form	Source of IL-1α	Study Type	Reference
Lung Cancer	Promote	Activates angiogenesis and lymphangiogenesis	Mature	Endogenous	In vitro and in vivo	[79]
Breast Cancer	Promote	Increases cell proliferation	Mature	Transduced	In vitro and in vivo	[49]
Breast Cancer	Promote	Induces TSLP expression from tumor-infiltrating myeloid cells to increase cancer survival and metastasis spread	Mature	Endogenous	In vitro, in vivo, and patient data	[80]
Breast Cancer	Promote	Increases activation of NF-kB and STAT3 to generate and maintain cancer stem cells.	Mature	Endogenous and Exogenous	In vitro, in vivo, and patient data	[52]
Breast Cancer	Promote	Associated with dedifferentiation and malignancy	Mature	Endogenous	In vitro and patient data	[53]
Breast Cancer	Suppress	Inhibits cell growth at G0/G1 phase	Mature	Exogenous	In vitro	[50]
Breast Cancer	Suppress	Suppresses cell proliferation through IL-1R signaling	Mature	Endogenous	In vitro and in vivo	[59]
Head and Neck Squamous Carcinoma	Promote	Associated with distant metastasis in patients	Mature	Endogenous	Patient data	[81]
Head and Neck Squamous Carcinoma	Suppress	Activates T-cell dependent anti-tumor response	Mature	Exogenous	In vitro, in vivo, and patient data	[82]
Liver Cancer	Promote	Activates inflammation and compensatory proliferation in liver	Mature	Endogenous	In vitro and in vivo	[83]
Liver Cancer	Suppress	Promotes T- and NK-cell activation	Membrane	Transduced	In vitro and in vivo	[84]
Pancreatic Cancer	Promote	Promotes HGF secretion by fibroblasts to promote cancer invasion, proliferation, and angiogenesis	Mature	Endogenous and Exogenous	In vitro	[60]
Pancreatic Cancer	Promote	Constitutively activates NF-kB to induce metastatic behavior	Mature	Transduced	In vitro and in vivo	[61]
Pancreatic Cancer	Promote	Sustains expression of inflammatory factors in tumor microenvironment beneficial for tumor survival	Mature	Endogenous	In vitro and patient data	[77]
Ovarian Cancer	Promote	IL-1α SNP (rs17561) associated with increase risk, possibly due to it being more readily cleaved to form mature IL-1α	Mature	Endogenous	Patient data	[85]
Acute Lymphocytic Leukemia	Promote	Facilitates growth of T-ALL cells through activation of NF-kB and SP1	N-Terminal	Transduced	In vitro and in vivo	[43]
Acute Lymphocytic Leukemia	Promote	Induces ALL cell growth in an IL-1α dose dependent manner	Mature	Endogenous and Exogenous	In vitro and patient data	[71]
Adult T-cell Leukemia	Promote	Induces ATL cell growth in an IL-1α dose dependent manner, possibly via autocrine mechanism	Mature	Exogenous	In vitro and patient data	[86]
Adult T-cell Leukemia	Promote	Associated with ATL patient samples and cell lines	Mature	Endogenous	In vitro and patient data	[87]
Adult T-cell Leukemia	Suppress	Stimulates anti-tumor immune responses	Mature	Endogenous	In vitro and in vivo	[88]
Fibrosarcoma	Promote	Involved in controlling the immune-surveillance of developing tumor	Mature	Endogenous	In vitro and in vivo	[89]
Fibrosarcoma	Suppress	Reduces tumorigenicity, increases immunogenicity, and regression of tumor	Cytosolic Full Length and Membrane	Endogenous and Transduced	In vitro and in vivo	[90]
Fibrosarcoma	Suppress	Increases immunogenicity, induces regression of tumor and development of systemic immunity	Cytosolic Full Length and Membrane	Endogenous and Transduced	In vitro and in vivo	[91]
Fibrosarcoma	Suppress	Stimulates anti-tumor immune responses and regression of tumor	Cytosolic Full Length and Membrane	Transduced	In vitro and in vivo	[92]
Gastric Cancer	Promote	Increases metastasis and tumor differentiation	Mature	Endogenous	Patient data	[83]
Gastric Cancer	Promote	Associated with enhance angiogenesis and metastasis	Mature	Endogenous	In vitro	[93]
Gastric Cancer	Promote	Associated with rapid progression to gastric pre-neoplasia	Full Length and Mature	Endogenous	In vitro and in vivo	[94]
Gastric Cancer	Promote	Positively correlated with carcinogenesis	Mature	Endogenous	In vitro and in vivo	[95]
Prostate Cancer	Promote	Correlated to increased serum PSA levels and progression of disease	Mature	Endogenous	Patient data	[96]
Oral Squamous Cell Carcinoma	Promote	Stimulates CAF proliferation and cytokine (CCL7, CXCL1, IL-8) secretion to promote OSCC cancer progression	Mature	Endogenous	In vitro, in vivo, and patient data	[97]
Cervical Cancer	Suppress	Activates dendritic cells to produce IL-12 for anti-tumor response	Mature	Endogenous	In vitro and patient data	[43]
Skin Cancer	Promote and Suppress	Suppresses carcinoma formation from prior papilloma.	Mature	Transduced	In vitro and in vivo	[98]
		Promotes carcinoma formation not from prior papilloma				
Skin Cancer	Promote	Induces adhesion molecules on endothelial cells, increasing tumor retention. As well as inducing potent inflammation, enhancing metastasis.	Mature	Transduced	In vitro and in vivo	[99]
Kidney Cancer	Promote	Induces malignant transformation	N-Terminal	Transduced	In vitro and in vivo	[75]
Multiple Cancers (T cell lymphoma, melanoma, lung carcinoma)	Suppress	Inhibits tumor growth by enhancing T-cell mediated antitumor immunity	Mature	Endogenous	In vitro and in vivo	[100]
Multiple Cancers (Various malignant human tumor cell lines)	Suppress	Induces cell apoptosis, possibly involving RNA processing apparatus	N-Terminal	Transduced	In vitro	[44]

Abbreviations: IL-1α, interleukin-1α; TSLP, thymic stromal lymphopoietin; NF- κB, nuclear factor kappa-B; STAT3, signal transducer and activator of transcription 3; HGF, hepatocyte growth factor; PSA, prostate specific antigen; CAF, cancer associated fibroblasts; CCL7, C-C motif chemokine ligand 7; CXCL1, C-X-C motif chemokine ligand 1; IL-8 interleukin -8; IL-12, interleukin-12.

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
