# Peer review of "IL-1α Processing, Signaling and Its Role in Cancer Progression"

_cells, 2021, doi:10.3390/cells10010092_

Round 1

Reviewer 1 Report

Comments:

IL-1α protein domain structure can be included. Which includes the NLS domain,

post-translational modification sites, and cleavage sites. 

Author Response

We thank the reviewer for the suggestions. We have included a schematic graph of IL-1α structure into Figure. 1. Please see the attachment. The figure legend is shown below:

Figure 1. Overview of IL-1α structure. Human IL-1α consists of 271 amino acids including a NLS domain. IL-1α can be cleaved by various proteases, such as Elastase, Granzyme B, Caspase-5, Thrombin, Chymase and Calpain. IL-1α also undergoes post-translational modification, including acetylation, myristoylation as well as phosphorylation.

Reviewer 2 Report

The review is clear, concise and well-written and can be accepted in its present form. 

Author Response

We thank the reviewer for the comments.

Reviewer 3 Report

This review reads well and is well structured. It is reasonably updated considering this field is very active and in this year we have already several papers which could have been included in. Worryingly enough, there is  another review recently published on IL1 and Cancer in one of our own MDPI journals:

Gelfo V, Romaniello D, Mazzeschi M, Sgarzi M, Grilli G, Morselli A, Manzan B, Rihawi K, Lauriola M. Roles of IL-1 in Cancer: From Tumor Progression to Resistance to Targeted Therapies. Int J Mol Sci. 2020 Aug 20;21(17):6009. doi: 10.3390/ijms21176009. PMID: 32825489; PMCID: PMC7503335.

Upon careful review both papers differ in their focus, Gelfo et al. is structured in terms of IL1 role in each hallmark whereas the present one is structured according to the role of IL1 in each cancer type.

I believe that I cannot fault the current one and will recommend acceptation but over to the editors to consider whether they retain this contribution original enough for their own journal after Gelfo et al.

Author Response

We thank the reviewer for the comments. We understand the concern that IL-1α has been extensively reviewed recently. However, this current manuscript is quite different from the one mentioned by the reviewer. 

1) The Gelfo's review focused on both IL-1α and IL-1β, while we only focus on IL-1α;

2) The Gelfo's review mainly described the role of IL-1 in carcinogenesis, while we also reviewed the structure, signaling, and subcellular localization of IL-1α;

3) We agree with the reviewer that Gelfo's review is structured in terms of IL-1 role in each hallmark whereas our review is structured according to the role of IL-1 in each cancer type. Furthermore, we also included the source and form of IL-1α in each study, which is of great importance as the functions of different form of IL-1α varies a lot.